EMBO
Molecular Medicine

# Human isotype-dependent inhibitory antibody responses against *Mycobacterium tuberculosis*

Natalie Zimmermann[1,2,3], Verena Thormann[1], Bo Hu[1], Anne-Britta Köhler[2], Aki Imai-Matsushima[4], Camille Locht[5,6,7,8,9], Eusondia Arnett[10], Larry S Schlesinger[10], Thomas Zoller[11], Mariana Schürmann[11], Stefan HE Kaufmann[2] & Hedda Wardemann[1,3,*]

## Abstract

Accumulating evidence from experimental animal models suggests that antibodies play a protective role against tuberculosis (TB). However, little is known about the antibodies generated upon *Mycobacterium tuberculosis* (MTB) exposure in humans. Here, we performed a molecular and functional characterization of the human B-cell response to MTB by generating recombinant monoclonal antibodies from single isolated B cells of untreated adult patients with acute pulmonary TB and from MTB-exposed healthcare workers. The data suggest that the acute plasmablast response to MTB originates from reactivated memory B cells and indicates a mucosal origin. Through functional analyses, we identified MTB inhibitory antibodies against mycobacterial antigens including virulence factors that play important roles in host cell infection. The inhibitory activity of anti-MTB antibodies was directly linked to their isotype. Monoclonal as well as purified serum IgA antibodies showed MTB blocking activity independently of Fc alpha receptor expression, whereas IgG antibodies promoted the host cell infection. Together, the data provide molecular insights into the human antibody response to MTB and may thereby facilitate the design of protective vaccination strategies.

**Keywords** antibodies; B cells; infection; isotype; *Mycobacterium tuberculosis*
**Subject Categories** Immunology; Microbiology, Virology & Host Pathogen Interaction

## Introduction

Tuberculosis (TB) is a major cause of death worldwide. Despite the extensive research, no effective vaccine against adult pulmonary TB has been developed (Andersen & Kaufmann, 2014). Therefore, long-term antibiotic treatment remains the standard of care (Zumla *et al*, 2013).

Stable *Mycobacterium tuberculosis* (MTB) infection is established in the lung after bacterial uptake by macrophages, which generally fail to eliminate the bacteria and instead serve as major MTB reservoir (Guirado *et al*, 2013). However, mycobacteria can also directly infect and replicate within epithelial cells, which has been associated with mycobacterial dissemination from the lung (Pethe *et al*, 2001; Castro-Garza *et al*, 2002). Surface-exposed MTB antigens play a major role in host cell invasion as shown, for example, for heparin-binding hemagglutinin (HBHA), an adhesion molecule that interacts with proteoglycans on the surface of epithelial cells, thereby facilitating MTB entry (Pethe *et al*, 2001; Locht *et al*, 2006; Menozzi *et al*, 2006). Moreover, lipids and lipoglycans, such as the abundant lipoarabinomannan (LAM), are major components of the MTB cell wall and promote the bacterial uptake through the interaction with mannose receptors expressed on the host cell surface (Schlesinger *et al*, 1994; Kang *et al*, 2005; Torrelles *et al*, 2008).

It is long known that T cells play a major role in TB immunity and are the target of current vaccination strategies (Cooper, 2009; Andersen & Kaufmann, 2014). However, more recent evidence points toward a role of B cells in modulating immune responses to MTB infection, for example, as antigen-presenting cells or through the production of cytokines and antibodies (Glatman-Freedman & Casadevall, 1998; Teitelbaum *et al*, 1998; Bosio *et al*, 2000; Chambers *et al*, 2004; Roy *et al*, 2005; Maglione *et al*, 2007, 2008; Achkar *et al*, 2015). Antibodies can exert their functions in two

1 Research Group Molecular Immunology, Max Planck Institute for Infection Biology, Berlin, Germany
2 Department of Immunology, Max Planck Institute for Infection Biology, Berlin, Germany
3 B Cell Immunology, German Cancer Research Center, Heidelberg, Germany
4 Department of Molecular Biology, Max Planck Institute for Infection Biology, Berlin, Germany
5 U1019 - UMR 8204 - CIIL - Centre for Infection and Immunity of Lille, University of Lille, Lille, France
6 CNRS, UMR 8204, Lille, France
7 Inserm, U1019, Lille, France
8 CHU Lille, Lille, France
9 Institut Pasteur de Lille, Lille, France
10 Center for Microbial Interface Biology, Department of Microbial Infection and Immunity, The Ohio State University, Columbus, OH, USA
11 Department of Infectious Diseases and Respiratory Medicine, Charité University Medical Center, Berlin, Germany
*Corresponding author. Tel: +49 6221 42 1270; Fax: +49 6221 42 1279; E-mail: h.wardemann@dkfz.de

ways: by the direct blocking of host cell invasion and neutralization of bacterial products, or indirectly through Fc-mediated complement and cell activation mechanisms through Fc receptors (Ravetch & Clynes, 1998). Steady-state mucosal immune responses are characterized by the production of IgA, whereas inflammatory conditions—including acute pulmonary TB—are associated with class switching to IgG (Brandtzaeg *et al*, 1997; Demkow *et al*, 2007).

Recent studies revealed that monoclonal antibodies against various mycobacterial surface antigens including LAM and HBHA can mediate protection in mouse models (Teitelbaum *et al*, 1998; Pethe *et al*, 2001; Chambers *et al*, 2004; Hamasur *et al*, 2004; Lopez *et al*, 2009). However, whether MTB-neutralizing antibodies against these structures also develop upon natural exposure in humans and induce the formation of B-cell memory remains to be determined. To address these questions, we analyzed the antibody repertoire of plasmablasts, a population of antibody-secreting B cells that circulate transiently in the blood upon acute infection and reflect the ongoing humoral immune response to pathogens, and the memory B-cell response to HBHA (Wrammert *et al*, 2008; Li *et al*, 2012). We demonstrate that MTB-exposed healthcare workers (HCW) and patients with active pulmonary TB generate B-cell antibody responses against mycobacterial surface antigens including LAM and HBHA, which mediate protection against the host cell invasion. Surprisingly, our data demonstrate that the inhibitory activity of anti-MTB antibodies depends directly on their isotype. IgA, but not IgG, antibodies specific for different MTB surface antigens can block MTB uptake by lung epithelial cells independently of the expression of IgA Fc receptors.

## Results

### Human anti-MTB plasmablast and memory B-cell antibody response

To characterize the antibody response to MTB, we initially screened peripheral blood samples from a cohort of 17 untreated and 8 drug-treated patients with active pulmonary TB for the presence of circulating plasmablasts and for serum antibody reactivity to MTB antigens (Fig 1). Healthy donors (HD) without contact with TB patients served as controls. About 50% of patients mounted specific IgG and IgA serum responses against MTB with higher antibody levels

against cell membrane antigens compared with secreted culture filtrate proteins (Fig 1A and B). Circulating $CD19^+CD27^{++}CD38^+$ plasmablasts were present in about 38% of all patients (Fig 1C). The cells expressed low levels of CD19 and were positive for the surface markers CD86, CD84, and CD24 indicative of their recent activation status (Fig 1D). The frequency of plasmablasts was highest during early acute TB and waned upon drug treatment, whereas serum IgG levels increased over six months of antibiotic combination therapy (Fig 1E). Several untreated patients with prominent serum antibody responses lacked the detectable plasmablast levels in the circulation, suggesting that these donors had been infected for prolonged times so that their circulating plasmablast response had waned (Fig 1F).

Next, we selected three patients with circulating plasmablasts and strong anti-MTB serum Ig responses for the molecular characterization of their plasmablast antibodies (TB7, TB24, TB33; Fig 2A; Appendix Table S1). From these donors, the *IGH* and corresponding *IGK* or *IGL* light chain transcripts of over 230 single isolated plasmablasts were amplified and sequenced (GenBank accession number KX947385–KX949063). To exclude any influence of the antibiotic drug treatment on our analyses, all samples were taken before the onset of therapy (Appendix Table S1). Consistently, the majority of TB plasmablasts in all donors expressed somatically mutated antibodies encoded by diverse Ig genes (Fig 2B; Appendix Table S2). MTB expresses a large number of diverse antigens. We therefore expected a high degree of polyclonality in the plasmablast response. Indeed, only a few cells from individual donors expressed Ig genes with identical heavy and light chain rearrangements as well as shared somatic mutations and thus were clonally related (GenBank accession number KX947385–KX949063). The relative bias toward IgA and near-complete absence of IgM expression compared with circulating memory B cells from the same donors indicated a mucosal origin (Fig 2C).

Plasmablasts can develop from naïve B cells or mutated memory B cells. The relatively high frequency of somatic mutations in plasmablasts at the levels comparable to circulating memory B cells under steady-state conditions suggested that the plasmablast response developed from reactivated memory B cells rather than from nonmutated naïve B cells that had been newly activated during active disease onset (Fig 2B; Tiller *et al*, 2007). To determine whether circulating anti-MTB memory B cells were detectable in the absence of active disease, we screened a cohort of healthy MTB-exposed HCW for the presence of circulating

---

**Figure 1.  Anti-MTB serum antibody response in the peripheral blood of TB patients.**

A    Serum IgG ELISA reactivity with MTB (H37Rv) whole-cell lysate, purified cell membrane fraction, and secreted culture filtrate proteins (CFP) for TB patients and healthy donors (HD). ELISA graphs (left; TB: black lines; *n* = 25; HD: red dotted lines; *n* = 2) and area under the curve (AUC) values for all tested samples (right; TB: *n* = 25; HD: *n* = 17) are shown. Median is shown.

B    AUC values for the IgA serum ELISA response against MTB whole-cell lysate, purified cell membrane fraction, and secreted culture filtrate proteins (CFP) of TB patients (*n* = 25) and healthy donors (HD; *n* = 9). Median is shown.

C    Flow cytometric gating strategy and frequency of circulating plasmablasts ($CD19^+CD27^{++}CD38^+$) for one representative TB patient (TB24) and HD (left). Frequency of circulating $CD19^+CD27^{++}CD38^+$ plasmablasts of all $CD19^+$ B cells in the peripheral blood of TB patients (*n* = 24) and HD (*n* = 8). Dashed line indicates the threshold for detectable plasmablast populations (right). Median and SEM are shown.

D    Expression of CD19 and activation markers (CD86, CD84, and CD24) in the plasmablasts and memory B cells as measured by flow cytometry.

E    Plasmablast response of patient TB7 (top) and IgG serum response against MTB cell lysates of patients TB7, TB19, and TB39 (bottom) and one HD at the indicated time points before and after the treatment onset.

F    Dots indicate AUC values for the anti-MTB cell lysate IgG serum ELISA response (*y*-axis) versus the frequency of circulating plasmablasts (*x*-axis) for individual TB patients. Spearman's correlation and corresponding *P*-value are shown.

Data information: All data are representative of two independent experiments. (A and B) *P*-values were determined using Mann–Whitney test; **P* < 0.01; *****P* < 0.0001; ns, not significant.

anti-MTB memory B cells. To identify MTB-reactive memory B cells among all circulating memory B cells, we focused our analysis on HBHA as representative MTB antigen and performed flow cytometry with fluorescently labeled HBHA to detect HBHA-reactive memory B cells in these donors compared with TB patients (Fig 2D and E). Although TB patients showed significantly

reduced overall memory B-cell frequencies compared with HD, we identified individual TB patients and MTB-exposed HCW with anti-HBHA memory B-cell responses (Fig 2E and F; Abreu *et al*, 2014). To determine the somatic mutation levels and the isotype distribution in these cells, we amplified and sequenced the Ig genes of single HBHA-reactive memory B cells from four HCW

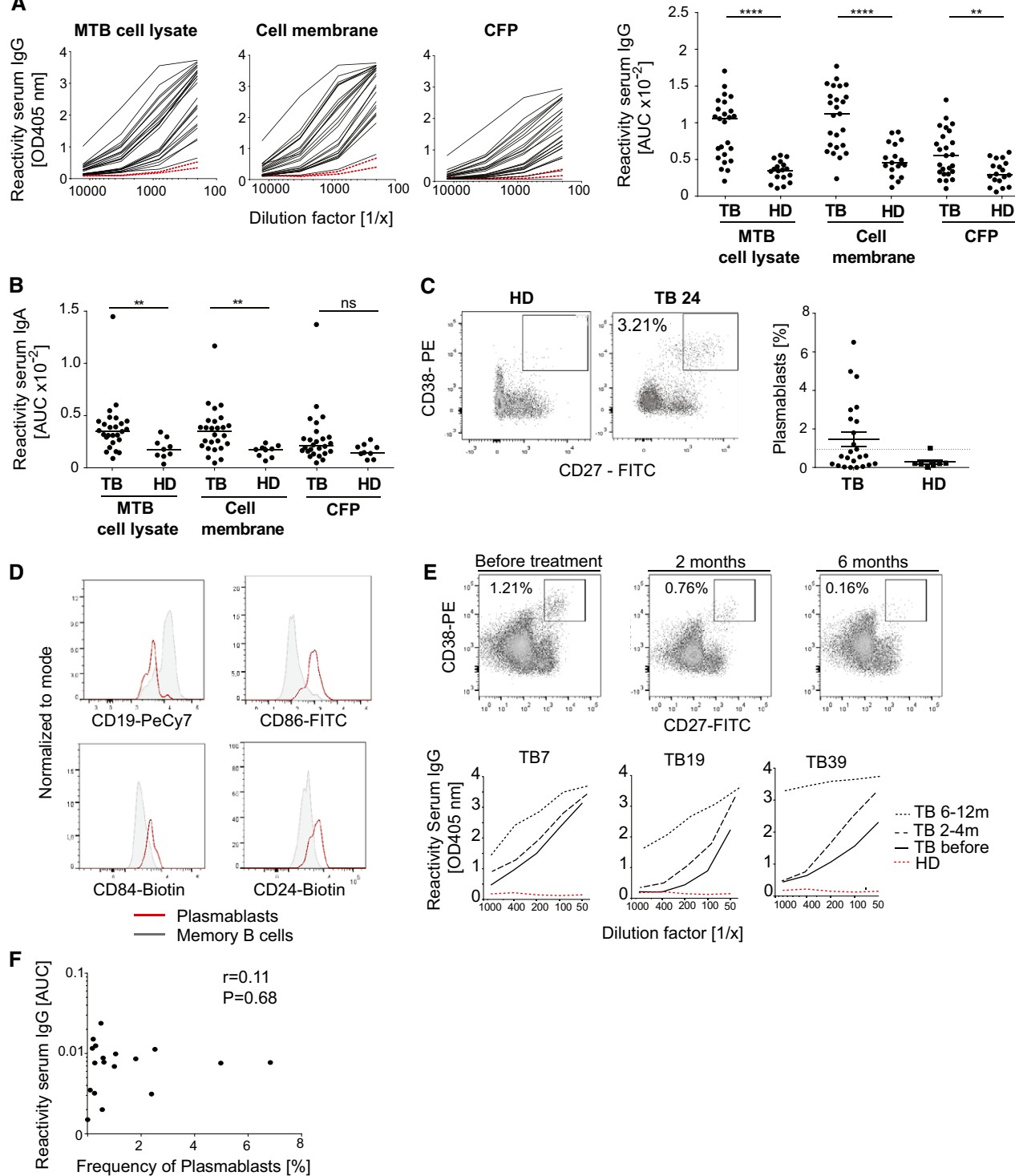

**Figure 1.**

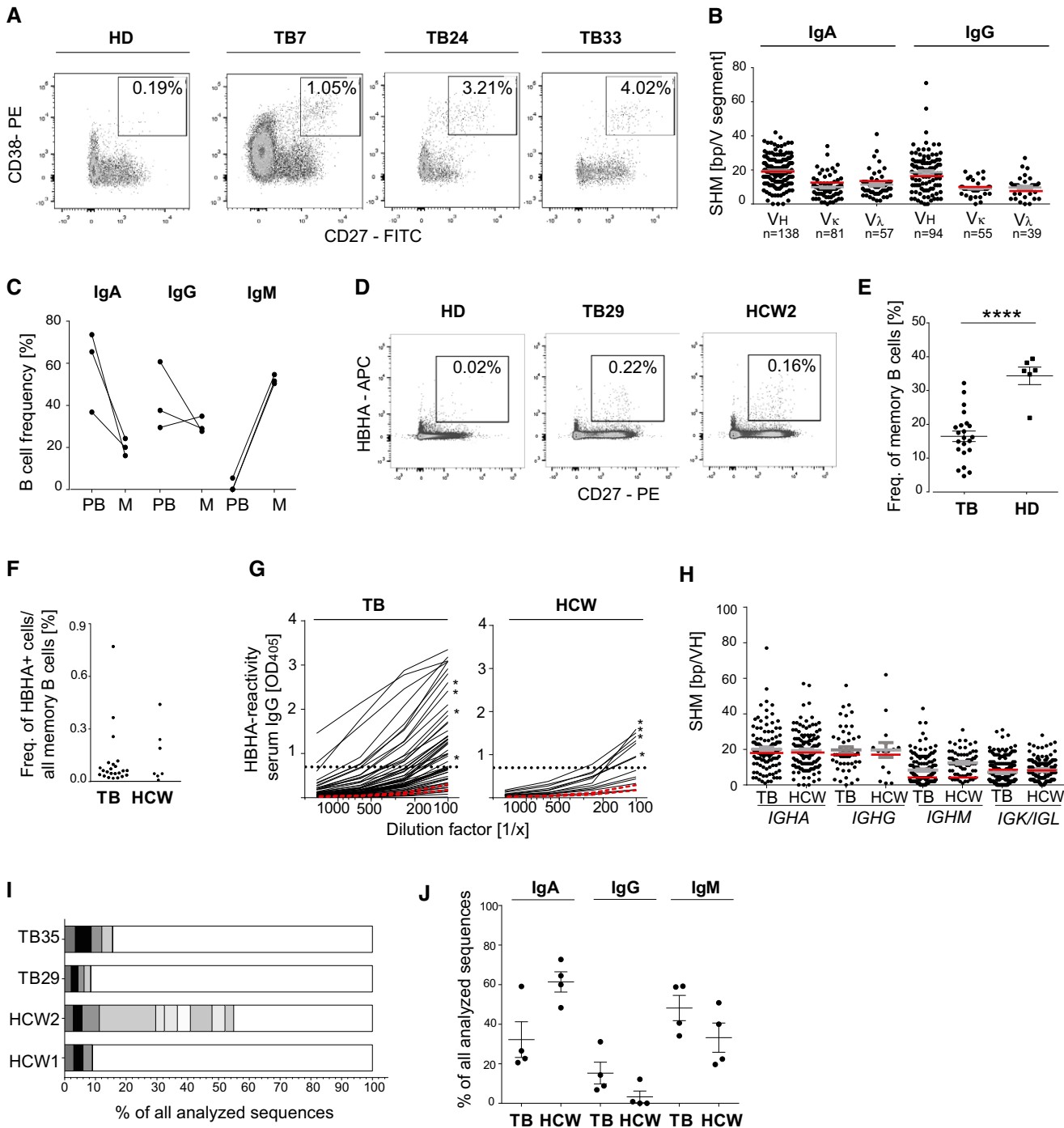

**Figure 2.**

and four TB patients with anti-HBHA serum titers (Fig 2G). Nearly all antibodies were somatically mutated at the levels comparable to the published data from unselected memory B cells of HD and we identified clusters of clonally expanded cells in all four donors (Fig 2H and I; Tsuiji *et al*, 2006; Tiller *et al*, 2007; Berkowska *et al*, 2015). *IGH* gene isotype analyses revealed a clear dominance of IgA and IgM over IgG anti-HBHA memory B-cell antibodies. The low frequency of IgG was more pronounced in HCW than in TB patients, whereas IgA was particularly more

abundant in HCW, suggesting an association of disease onset with the induction of IgG responses (Fig 2J).

In summary, the data provide evidence that circulating plasmablasts in the peripheral blood of patients with active pulmonary TB develop from a polyclonal set of mutated and reactivated memory B cells. The high frequency of IgA anti-HBHA memory B cells in HCW suggests that memory is formed upon primary MTB exposure presumably from mucosal immune responses. Active TB could lead to the reactivation of preexisting memory B cells and the

◄

**Figure 2.   Somatic hypermutation level and isotype distribution of single-cell-sorted plasmablasts and antigen-specific memory B cells.**

A    Gating strategy, phenotype, and frequency of circulating plasmablasts (CD19$^+$CD27$^{++}$CD38$^+$) isolated by flow cytometric cell sorting from three TB patients (TB7, TB24, and TB33) in comparison with one representative HD. Boxes indicate sort gates. The plasmablast frequency is indicated.

B    Absolute number of somatic hypermutations (SHM) in the *IGHV*, *IGKV*, and *IGLV* segments of IgA and IgG plasmablast antibody genes sequenced from TB7, TB24, and TB33. The absolute number of sequences analyzed is indicated below the graph. Geometric means with SEM are indicated in gray. SHM means of historic data from sorted CD27$^+$IgA$^+$ or CD27$^+$IgG$^+$ cells from the peripheral blood of HD are indicated in red for comparison (Tiller *et al*, 2007; Berkowska *et al*, 2015).

C    Isotype distribution of plasmablast and memory B cells from TB7, TB24, and TB33. PB, plasmablasts (CD19$^+$CD27$^{++}$CD38$^+$); M, memory B cells (CD19$^+$CD27$^+$).

D    Gating strategy, phenotype, and frequency of HBHA-reactive memory B cells (CD19$^+$CD27$^+$HBHA$^+$) in the peripheral blood of one representative TB patient (TB29), healthcare worker (HCW) 2, and HD, respectively.

E    Dots indicate the frequency of HBHA-reactive memory B cells out of all CD27$^+$ memory B cells in individual TB patients (*n* = 23) and HCW (*n* = 7). Mean and SEM are indicated. *P*-value was determined using Wilcoxon–Mann–Whitney test; ****$P < 0.0001$.

F    Dots indicate the frequency of resting memory B cells (CD19$^+$CD27$^+$CD10$^-$) out of all B cells in the peripheral blood of individual TB patients compared with HCW.

G    Anti-HBHA serum IgG ELISA reactivity for TB patients and HCW (black lines) compared with two representative HDs (red lines). Dashed line indicates the threshold OD$_{405\ nm}$ for positive reactivity. Asterisks indicate the serum responses of donors selected for single sorting of HBHA-reactive memory B cells. Data are representative of two independent experiments.

H    Absolute number of somatic hypermutations (SHM) in the IGHV (*IGHA*, *IGHG*, and *IGHM*), *IGKV*, or *IGLV* segments of sorted anti-HBHA memory cells from TB patients and HCW. Geometric means with SEM are indicated in gray. For comparison, red lines indicate the historic SHM means for randomly sorted CD27$^+$IgA$^+$, CD27$^+$IgG$^+$, or CD27$^+$IgM$^+$ cells from the peripheral blood of HDs (Tsuiji *et al*, 2006; Tiller *et al*, 2007; Berkowska *et al*, 2015).

I     The number and size of clonally expanded B-cell clusters among HBHA-reactive memory B cells from two TB patients (TB35 and TB29) and HCW (HCW1 and HCW2). Cells in clusters are indicated in gray, and single cells are indicated in white. No B-cell clusters were shared between donors.

J     IgA, IgG, and IgM isotype distribution of single-cell-sorted HBHA-reactive memory cells. Mean and SEM are indicated.

formation of plasmablast responses that are associated with class switching to IgG.

**Plasmablast antibodies frequently target MTB surface antigens**

Antibodies targeting surface-exposed bacterial antigens likely play a functional role in the anti-MTB response. To determine whether the B-cell response to MTB produces functional antibodies, we cloned the *IGH* and corresponding *IGK* or *IGL* genes from 113 IgA$^+$ and IgG$^+$ plasmablasts and produced the recombinant monoclonal antibodies *in vitro* (Appendix Table S2). All antibodies were initially produced as IgG1 to allow for the direct comparison of their antigen-binding capacity independently of the original plasmablast isotype. We then tested the antibodies for binding to MTB cell lysate or whole bacteria by ELISA (Fig 3A and B). On average, 40% of all recombinant monoclonal antibodies were MTB reactive in these assays (Fig 3C). To identify nonspecific binding of antibodies, we also tested all antibodies for binding to irrelevant and structurally diverse antigens (dsDNA, insulin, LPS). Cross-reactivity was detected for about 16% of antibodies, indicating that the majority of plasmablast antibodies were antigen specific (Appendix Table S2).

A large fraction of anti-MTB antibodies recognized whole MTB bacteria in the ELISA, suggesting that they may target bacterial surface antigens (Fig 3B). We therefore tested a selected set of MTB-reactive and nonreactive antibodies for binding to cell membrane antigens by ELISA (Fig 3D). Indeed, the majority (57.6%, 15/26) of anti-MTB but only 1 of 15 nonreactive control antibodies was reactive with purified cell membrane antigens (Appendix Table S2). For individual antibodies, binding to the mycobacterial surface was confirmed by flow cytometry, suggesting that they recognized epitopes that are accessible to antibodies *in vivo* (Fig 3E and Appendix Table S2). Thus, the human plasmablast antibody response to MTB infection predominantly targets mycobacterial surface antigens.

LAM is a major component of the mycobacterial cell surface and a target of protective antibodies (Brown *et al*, 2003; Hamasur *et al*, 2004). We therefore interrogated whether LAM served as target antigen of human anti-MTB plasmablast antibodies (Fig 4A–C;

Appendix Table S2). By Western blot, FACS, and ELISA, we identified two antibodies that recognize MTB ManLAM. Antibody TB24PB037 lacked reactivity with PILAM from the nonpathogenic *Mycobacterium smegmatis*, demonstrating its high specificity for MTB compared with the CS-35-positive control antibody (Fig 4D).

HBHA represents another surface antigen and MTB virulence factor, which is targeted by antibodies (Menozzi *et al*, 1996; Pethe *et al*, 2001; Kohama *et al*, 2008). Although none of the plasmablast antibodies recognized HBHA (data not shown), the detection of HBHA-reactive memory B cells in HCW and TB patients with acute disease (Fig 2D and F) prompted us to determine whether these cells expressed HBHA-specific antibodies (Fig 4E–G). Upon Ig gene cloning and *in vitro* expression of antibodies from HBHA-reactive memory B cells, we identified 25 HBHA binders by ELISA or Western blot. Thirteen antibodies of all isotypes lacked cross-reactivity and were therefore HBHA specific (Appendix Table S3). Binding of HBHA-reactive antibodies to the outer surface of mycobacteria was confirmed by fluorescence microscopy (Fig 4G).

Thus, we conclude that MTB-exposed individuals mount high-affinity plasmablast and memory antibody responses against MTB surface antigens such as ManLAM and HBHA relevant for host cell infection.

**Antibody isotype-dependent functional differences in MTB inhibition**

To determine the potential role of both plasmablast and memory B-cell antibodies in MTB infection, we tested a selected set of 41 MTB-reactive recombinant monoclonal antibodies in an *in vitro* infection assay with A549 human lung epithelial cells (Fig 5 and Appendix Table S4). A549 cells are type II alveolar epithelial cells that have been shown to play a role in early MTB infection (Bermudez & Goodman, 1996; Castro-Garza *et al*, 2002; Sato *et al*, 2002; Ryndak *et al*, 2015; Zimmermann *et al*, 2016). We initially expressed and tested a set of antibodies that had been cloned from IgA-positive plasmablasts including the anti-LAM antibody TB24PB037 (Fig 5A and B). To mimic the original plasmablast and memory B-cell isotype and subclass, we expressed the antibodies as

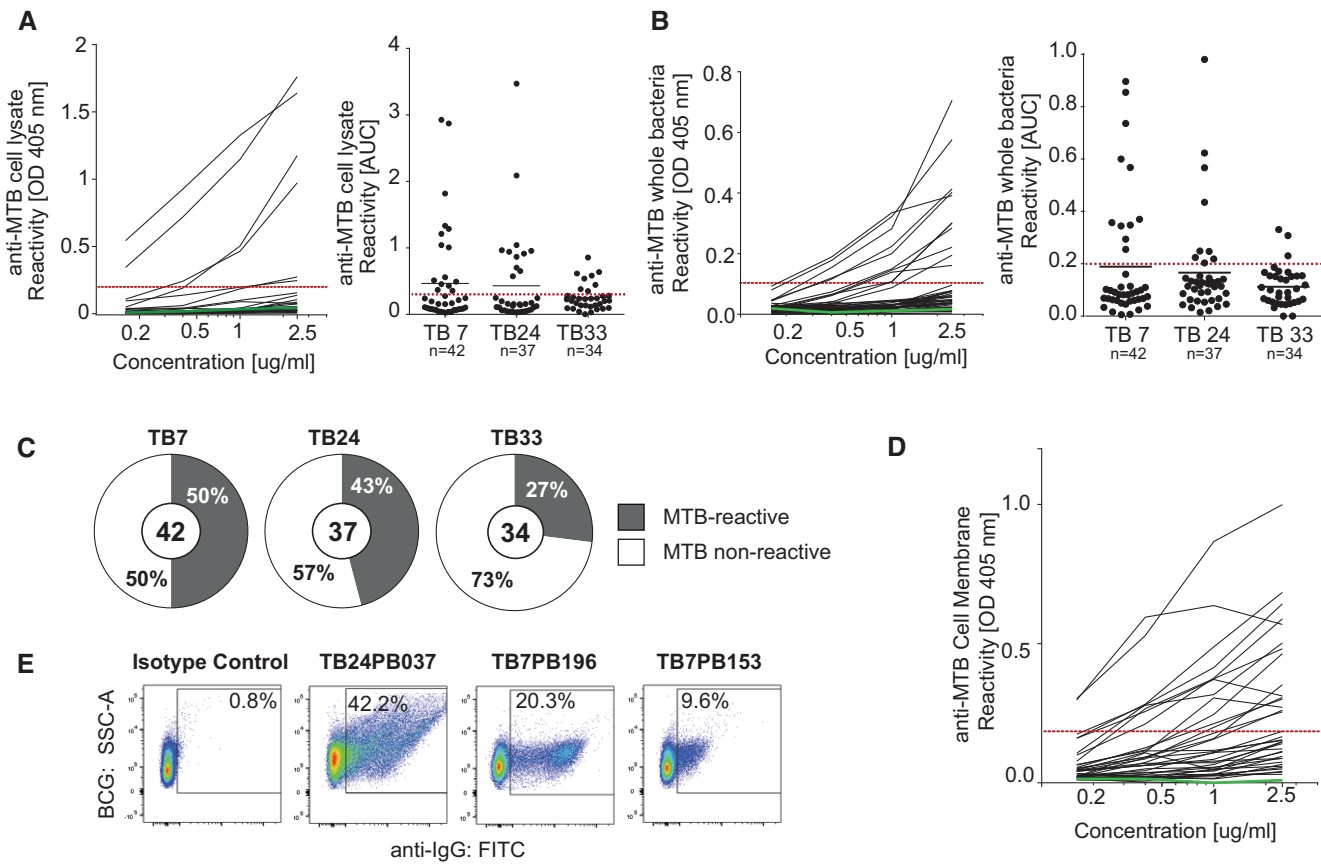

**Figure 3. Peripheral plasmablast antibodies from TB patients bind to mycobacterial surface antigens.**

A total of 113 recombinant monoclonal plasmablast antibodies were generated and characterized for antimycobacterial reactivity.

A, B Representative ELISA graphs show the reactivity of antibodies from patient TB7 to (A) MTB whole-cell lysate or (B) MTB bacteria (left). Dashed red line indicates the threshold $OD_{405\ nm}$ for positive reactivity. Green line indicates the negative control antibody (mGO53; Wardemann *et al*, 2003). AUC (area under curve) values indicate the reactivity to (A) MTB cell lysate or (B) whole MTB bacteria for antibodies from TB7, TB24, and TB33. The total numbers of analyzed antibodies are indicated (right).

C Pie charts show the frequency of MTB-reactive (gray) vs. nonreactive (white) antibodies as measured by whole-cell lysate and MTB bacteria ELISA for each patient. MTB-reactive antibodies were positive in both or one of the assays. The total numbers of analyzed antibodies are indicated in the center of the charts.

D ELISA graph shows the reactivity to the MTB cell membrane fraction for MTB-reactive (*n* = 26) and nonreactive (*n* = 14) antibodies.

E Reactivity to whole BCG bacteria as determined by flow cytometry for three representative MTB cell lysate-reactive antibodies and one nonreactive antibody (isotype control, mGO53).

Data information: All data in (A, B, D, E) are representative of at least two independent experiments.

IgA1 or IgA2, respectively (Appendix Table S2; Lorin & Mouquet, 2015). For TB24PB037 and six additional plasmablast antibodies of unknown antigen specificity, preincubation of the mycobacteria reduced the intracellular bacterial load by 20–50% compared with the medium control, whereas 5/12 IgA plasmablast antibodies and the non-MTB-reactive isotype control showed no effect (Fig 5A and B). Inhibition was not associated with a specific IgA subclass, suggesting that the differences in the functional properties of individual plasmablast antibodies can be explained by the differences in the target antigen/epitope specificity or affinity. Antibodies without effect may target antigens or epitopes irrelevant for infection.

To determine whether IgG plasmablast antibodies showed similar inhibitory activities for MTB, we tested 26 MTB-reactive IgG antibodies that had been cloned from IgG$^+$ plasmablasts in the same assay (Fig 5A and B). Surprisingly, none of the IgG antibodies was able to inhibit infection. Instead, most monoclonal anti-MTB IgG

plasmablast antibodies increased the mycobacterial load (Fig 5B). The results were confirmed with anti-HBHA memory B-cell IgA and IgG antibodies (Fig 5C and D). Whereas IgA anti-HBHA monoclonal antibodies impaired the infection, HBHA-reactive IgG antibodies promoted the infection or had no effect.

The observation that IgA antibodies inhibited the infection of epithelial cells whereas IgG antibodies promoted mycobacterial invasion suggested that the differences in isotype rather than antigen or epitope specificity were associated with functional differences in MTB inhibitory activity. We therefore cloned and directly compared IgG and IgA versions of 10 MTB-reactive antibodies including the anti-LAM antibodies (Fig 5E). Indeed, for all tested antibodies, IgG increased, while IgA decreased the bacterial load. Antigen binding was not influenced by antibody isotype as measured by ELISA and the effect was concentration dependent (Appendix Fig S1).

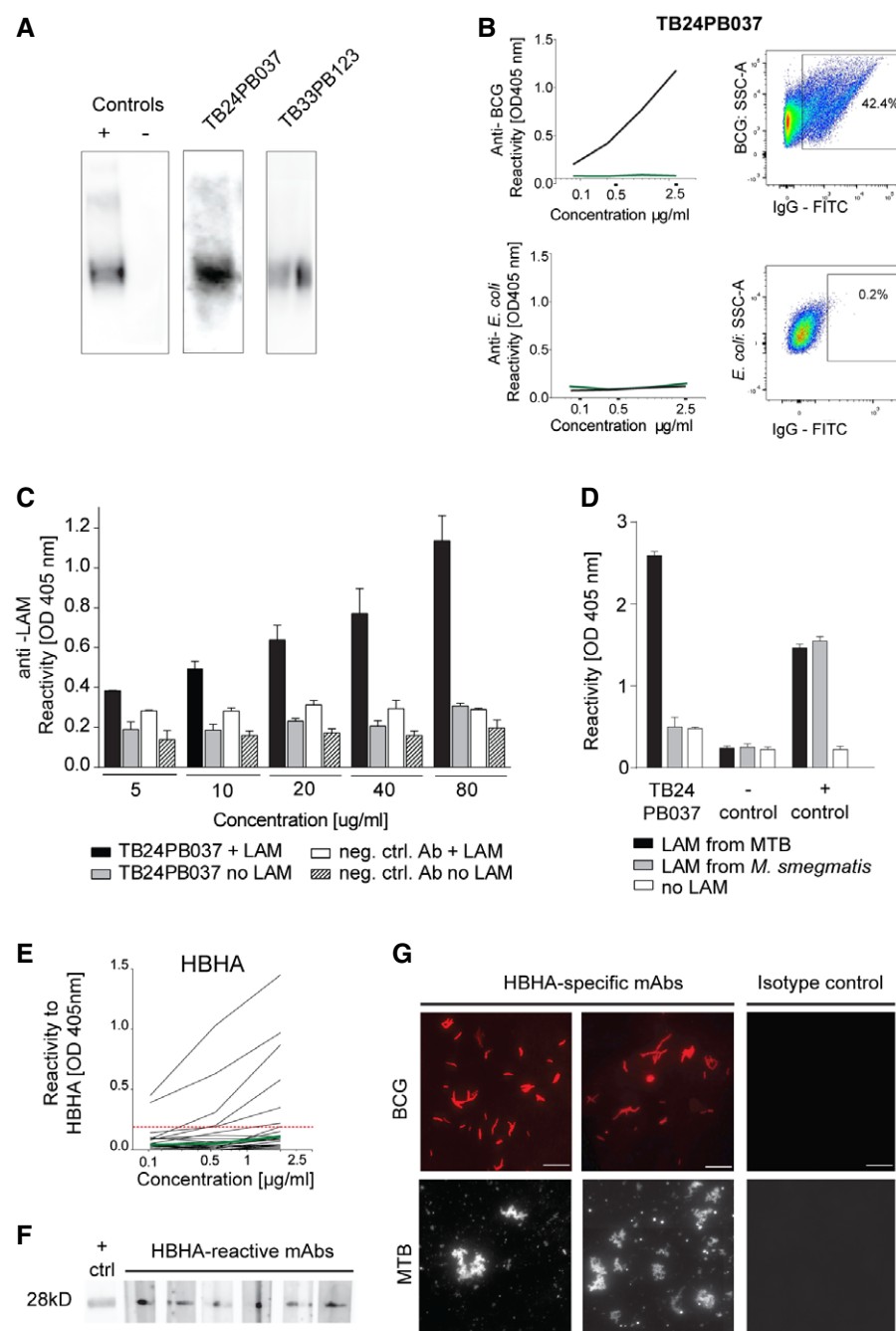

**Figure 4. The surface-exposed virulence factors LAM and HBHA are targets of the human plasmablast and memory B-cell response to MTB.**

A   Binding of antibodies TB24PB037 and TB33PB123 to LAM purified from MTB as determined by Western blot. The negative isotype control (mGO53) and a commercially available anti-LAM antibody (positive control) are shown for comparison.

B   Anti-BCG specificity of the LAM-reactive antibody TB24PB037 (black) compared with *E. coli* as determined by ELISA with whole bacteria (left) and flow cytometry (right).

C   Anti-LAM ELISA for TB24PB037 or negative control antibody at the indicated concentrations. The mean ± SD of the absorbance was calculated.

D   Antibody TB24PB037 binds to MTB-LAM, but not to LAM from *M. smegmatis*. ELISA performed as described in (C). The mean ± SD of the absorbance was calculated.

E   Representative HBHA ELISA for antibodies cloned from HBHA-reactive memory B cells of patient TB35 and HCW1. Dashed red line indicates the threshold $OD_{405\,nm}$ for positive reactivity. Green line indicates the negative control antibody (mGO53; Wardemann *et al*, 2003).

F   Reactivity to HBHA was confirmed by Western blot for a selected set of antibodies with HBHA ELISA reactivity.

G   Fluorescence microscopy shows BCG and MTB reactivity of representative anti-HBHA antibodies compared with a nonreactive isotype control antibody. Scale bars, 10 μm.

Data information: All data are representative of two independent experiments.
Source data are available online for this figure.

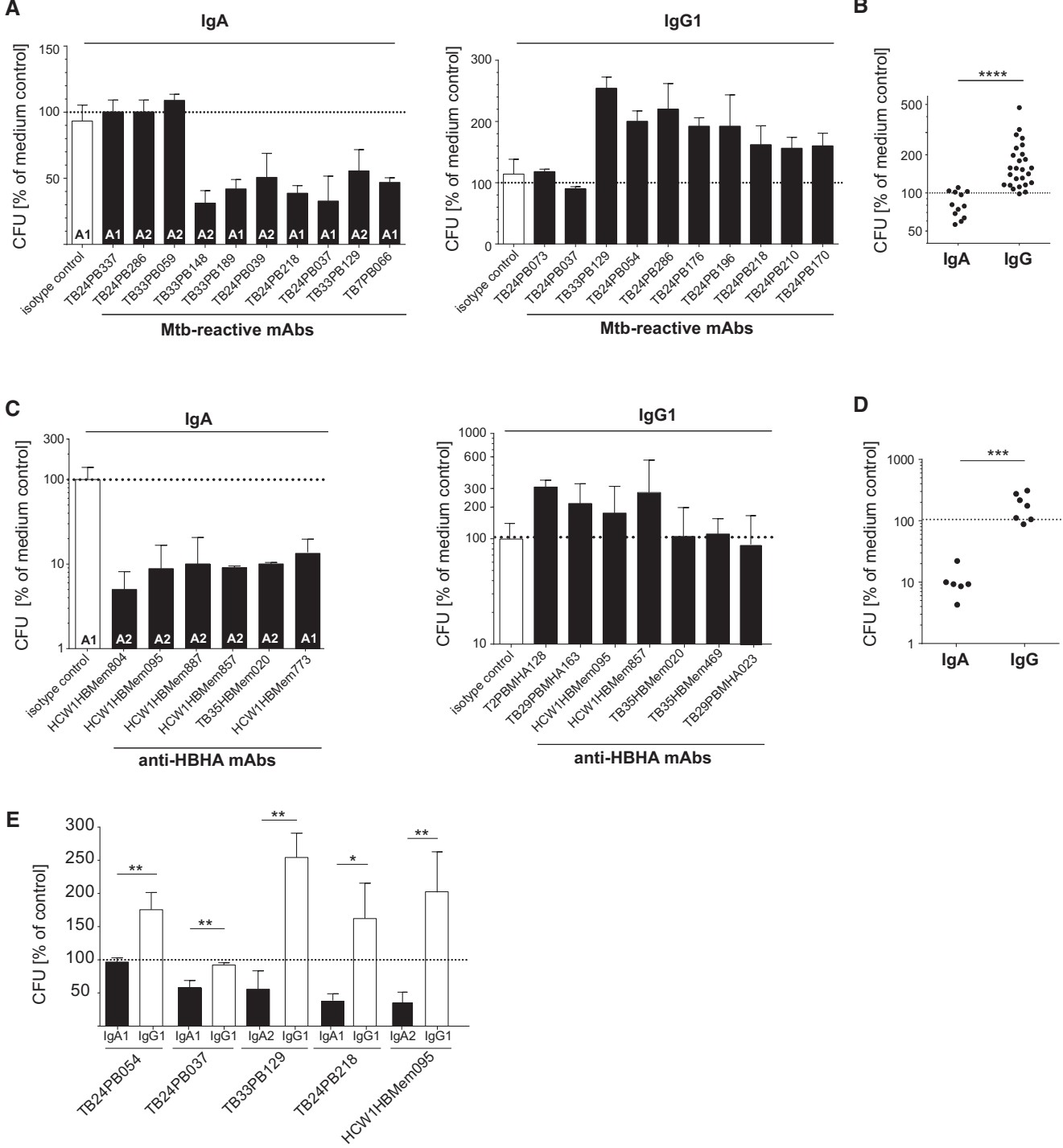

**Figure 5. IgA and IgG isotype-dependent functional differences in the effect of monoclonal antibodies on MTB infection of human lung epithelial cells.**

Relative bacterial counts (CFU) compared with the medium control (100%; dashed lines) in human lung epithelial A549 cells 1 h after infection with MTB, preincubated with recombinant MTB-reactive IgA1, IgA2, or IgG1 antibodies and nonreactive isotype control as indicated.

A   Representative results for individual MTB-reactive plasmablast antibodies. Mean and SEM are indicated.
B   Direct comparison of all tested MTB-reactive IgA (*n* = 12) and IgG (*n* = 26) antibodies.
C   Representative results for individual HBHA-reactive memory B-cell antibodies from TB patients and HCW.
D   Same as in (B) but with HBHA-reactive antibodies as shown in (C). Direct comparison of all tested HBHA-reactive IgA (*n* = 6) and IgG (*n* = 7) antibodies.
E   Direct comparison of the indicated MTB-reactive antibodies expressed as IgA and IgG. Mean and SEM are indicated.

Data information: Data are representative of at least two independent experiments with three technical replicates each. (B, D) *P*-values were determined using Wilcoxon–Mann–Whitney test, \*\*\**P* < 0.001; \*\*\*\**P* < 0.0001. (E) *P*-values were determined using nonparametric *t*-test, \**P* < 0.1; \*\**P* < 0.01, ns: *P* > 0.5.

To determine whether the differential expression of Fc receptors may be underlying the isotype-dependent differences in antibody-mediated inhibition, we used flow cytometry and PCR to measure the IgG and IgA Fc receptor expression levels of A549 and primary human airway epithelial cells (Fig 6A and B). Compared with THP-1 macrophages as a positive control, A549 and primary human airway epithelial cells did not express the IgA Fc receptor CD89 or conventional IgG Fc receptors, which promote the bacterial uptake through IgG immune complexes (Fig 6A). However, epithelial cells were positive for the neonatal Fc receptor, which efficiently binds IgG and could thereby promote cell infection (Fig 6B; Spiekermann *et al*, 2002). Despite the difference in CD89 and FcγR expression, infection of macrophages with anti-LAM (TB24PB037) IgA- or IgG-coated MTB (Fig 6C) reproduced our findings in the epithelial cells (Fig 5). Preincubation with IgA reduced the bacterial load, whereas IgG did not. We conclude that the IgA-mediated blocking effect of MTB uptake (Figs 5 and 6C) was not IgA Fc receptor mediated and uptake of IgG-coated bacteria was not influenced by FcγR but more likely by FcRn expression.

To determine whether isotype-associated differences in mycobacterial uptake were also observed for serum antibodies, we purified serum IgA and IgG from TB patients and non-TB-exposed HDs

(Fig 7). Increased uptake of MTB by A549 cells was detected for purified polyclonal IgG preparations from the serum of individual TB patients, whereas purified serum IgA reduced the mycobacterial load similar to the monoclonal plasmablast antibodies (Fig 7A and B). In summary, our data suggest that isotype plays a critical role in determining the inhibitory vs. noninhibitory properties at the monoclonal and polyclonal serum antibody level in TB.

## Discussion

The role of humoral immune responses in TB is still highly controversial (Achkar *et al*, 2015). We here demonstrate that the plasmablast antibody response to MTB infection in humans generates functionally active antibodies. The expression of high-binding somatically mutated and class-switched antibodies suggests that these antibodies underwent multiple rounds of selection to reach higher affinity and most likely developed with the help of T cells in germinal center reactions. In low-incidence countries, the vast majority of active TB cases are believed to be the consequence of disease reactivation in individuals with latent TB infection, who failed to contain MTB within solid granulomas (Small *et al*, 1994).

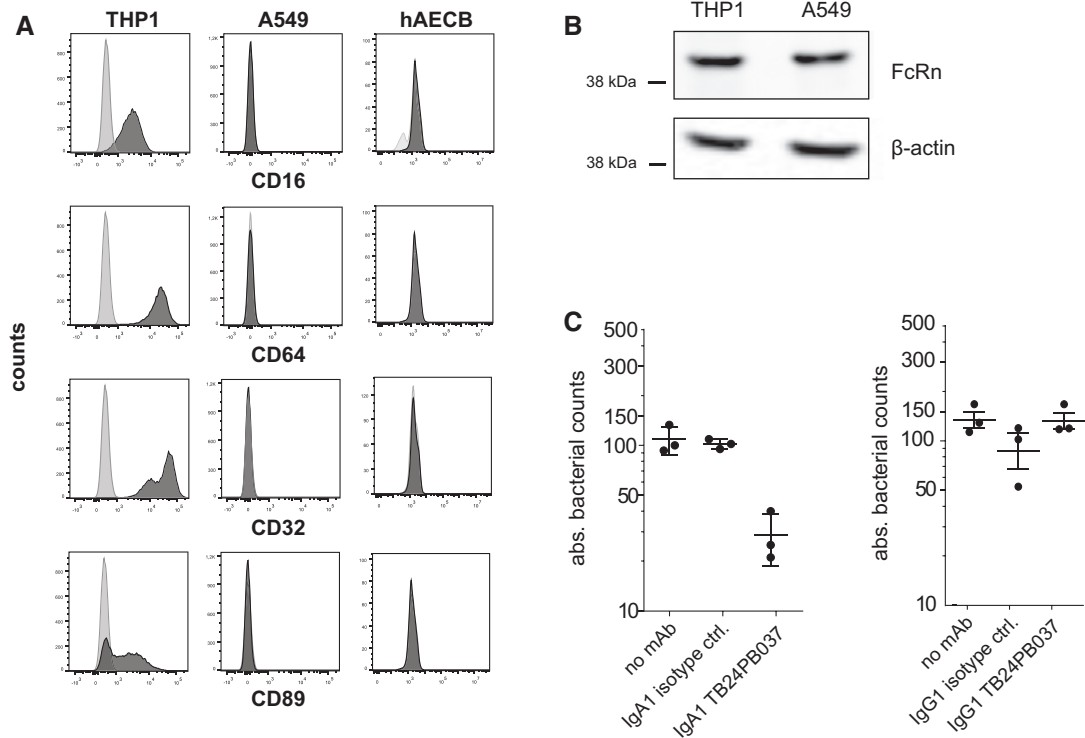

**Figure 6.  Fc receptor expression on human lung epithelial cells.**

A   Human lung epithelial A549 cells and primary bronchial epithelial cells (hAECB) lack FcγR (CD16, CD64, and CD32) and FcαR (CD89) cell surface expression as analyzed by flow cytometry. Human THP1 cells serve as a positive control.

B   The neonatal FcR (FcRn) is expressed in both A549 cells and human THP1 cells as determined by Western blotting. Data are representative of at least two independent experiments.

C   Absolute bacterial counts (CFUs) in human THP1 macrophages 1 h after infection with MTB preincubated with the LAM-reactive IgA1 (left) or IgG1 (right) antibody TB24PB037 or the respective isotype control antibody as indicated. Mean and SEM are indicated.

Data information: Data are representative of three independent experiments.

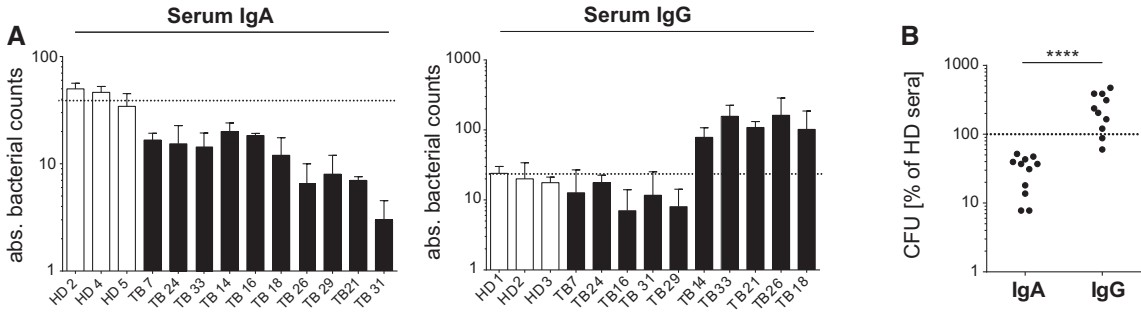

**Figure 7.  IgA and IgG isotype-associated differences in the effect of TB serum antibodies on MTB infection of human lung epithelial cells.**

A   Absolute bacterial counts (CFUs) in human lung epithelial A549 cells 1 h after infection with MTB preincubated with purified serum IgA (left) or IgG (right) antibodies from TB patients (TB) compared with HD as indicated. Mean and SEM are indicated. Data are representative of three independent experiments with three technical replicates each.

B   Direct comparison of purified serum IgA and IgG antibodies on epithelial cell invasion as measured in (A). *P*-value was determined using Wilcoxon–Mann–Whitney test; ****P* < 0.0001.

Data information: Data are representative of two independent experiments with three technical replicates each. Dotted lines indicate the medium control.

Thus, it is likely that the plasmablast response studied here originated from reactivated memory B cells that were generated during local immune responses to primary MTB infection. Indeed, proliferating and nonproliferating IgA$^{+}$ plasma cells as well as small numbers of IgG plasma cells could be detected adjacent to caseous necrotic granulomas in the lung sections of TB patients with active disease (Fig EV1). The clear bias toward IgA and the fact that MTB-specific IgA and IgG memory B cells can be isolated from the peripheral blood of healthy TB-exposed HCW—as demonstrated for anti-HBHA memory B cells—suggests that the anti-MTB B-cell response spreads from the lung to the periphery independently of active TB disease.

Our data demonstrate that a substantial fraction of the plasmablast response is directed against cell membrane antigens including ManLAM, a target of protective antibodies (Brown *et al*, 2003). Thus, the targeted induction of high-affinity antibodies and memory B cells against MTB surface antigens such as ManLAM or HBHA could protect against primary infection prior to MTB entry into the lung parenchyma and the establishment of stable infection, for example, by inhibiting the interaction with the mannose receptor on macrophages or through binding to proteoglycans on the surface of epithelial cells, respectively (Teitelbaum *et al*, 1998; Pethe *et al*, 2001; Hamasur *et al*, 2004; Kang *et al*, 2005). Although we did not detect differences in the specificity of antibodies cloned from different donors, it is likely that the target antigen specificity and the affinity of memory B cell, plasmablast, and total serum antibodies play a major role in determining the overall functionality of the humoral immune response. However, antibody cross-reactivity does not seem to influence the inhibitory activity of anti-MTB antibodies.

Airway responses are typically IgA biased, and passive immunization with IgA can protect against early MTB infection of the lung (Brandtzaeg *et al*, 1997; Rojas & Apodaca, 2002; Williams *et al*, 2004; Buccheri *et al*, 2007; Balu *et al*, 2011). However, IgG responses develop particularly under inflammatory conditions and anti-MTB IgG serum levels are higher in patients with active disease compared with HCW (Demkow *et al*, 2007; Pukazhvanthen *et al*, 2014). Differences in antigen specificity and/or affinity or low anti-MTB antibody titers may explain the lack of any inhibitory or

enhancing effect for individual monoclonal antibodies and serum IgA or IgG preparations from individual donors, respectively. How antibody isotype differences influence the course of natural MTB infection in humans and whether isotype switching precedes the development of active TB or reflects a consequence of disease remain to be determined, but our data suggest that the expression of IgG may promote MTB uptake by alveolar epithelial cells.

Experiments in mice have demonstrated a protective role for antibodies in certain infection models. For example, intranasal administration of MTB together with a mouse monoclonal anti-HBHA IgG delayed bacterial dissemination to the spleen and liver, without affecting the bacterial load in the lung (Pethe *et al*, 2001). In contrast, intravenous infection and administration of a LAM-specific murine IgG led to a reduction in the bacterial load in the lung and was associated with the prolonged survival (Hamasur *et al*, 2004). Collectively, these published data suggest that the protective function of anti-MTB antibodies, at least in part, may depend on the target antigen and host cell type, the local environment, and the route of administration (mucosa vs. periphery; Falero-Diaz *et al*, 2000; Pethe *et al*, 2001; Chambers *et al*, 2004; Hamasur *et al*, 2004; Williams *et al*, 2004; Roy *et al*, 2005; Lopez *et al*, 2009). However, no systematic comparison of isotype-associated functional differences has been made previously. Our data suggest that antibody isotype differences in preexisting memory B cells and plasma cells together with the differences in the antigen/ epitope specificity of the antibody response may play a major role in modulating the course of the infection and the development and progression of disease in humans. The targeted induction of IgA memory B cells and serum antibodies may help to protect from MTB infection or promote immunity upon primary exposure. Due to the incomplete Fc receptor homology and the differences in expression patterns between mice and humans, mouse models may be insufficient to assess the isotype-associated functional differences for human antibodies *in vivo* even if chimeric antibody constructs with murine Fc parts were generated (Bournazos *et al*, 2015). The development of mice expressing human Fc receptors may be necessary to overcome these limitations (van Egmond *et al*, 1999; Bakema & van Egmond, 2011; Smith *et al*, 2012). Thus, a better

understanding of the mechanisms underlying the development, induction, and function of IgA vs. IgG responses in humans as well as the identification of protective MTB target epitopes can form the basis for the rational design of a vaccine, which not only prevents TB disease, but also stable MTB infection.

# Materials and Methods

### Study participants

Patients included in this study were 19–57 years of age (mean $42 \pm 10$ years). Active pulmonary TB was diagnosed by the presence of typical clinical symptoms in conjunction with a chest X-ray compatible with TB as well as the diagnosis of MTB by sputum smear microscopy and/or positive culture. After signed informed consent was obtained, peripheral blood was drawn before or after the start of therapy, as indicated.

Healthcare workers (HCW) who have been working with TB patients for several years were recruited at the Charité Hospital, Berlin.

Peripheral blood from healthy donors (HD) with no history of MTB infection and no reported contact with TB patients served as controls. Peripheral blood mononuclear cells and plasma were purified using Ficoll gradient density centrifugation.

### Bacteria, bacterial fractions, antigens, and cell lines

All assays were performed using the MTB strain H37Rv. For the assessment of monoclonal antibody binding to mycobacteria by flow cytometry, the attenuated vaccine strain *M. bovis* Bacillus Calmette–Guérin (BCG; Denmark background SSI) was used. Bacteria were grown in Middlebrook 7H9 broth supplemented with Middlebrook ADC, 0.05% Tween-80, and 0.5% glycerol, or on Middlebrook 7H11 agar plates supplemented with Middlebrook OADC and 0.05% Tween-80. Before the host cell infection, bacteria were diluted to a multiplicity of infection (MOI) of 1 and passaged five times through a tuberculin syringe to obtain the single-cell suspensions. Unless indicated otherwise, all mycobacterial fractions including the H37Rv cell membrane fraction (NR-14831), H37Rv culture filtrate proteins (NR-14825), and all purified mycobacterial antigens including the H37Rv LAM (NR-14848) were obtained from BEI Resources, NIAID, NIH.

Native HBHA was expressed and purified from *M. bovis* BCG by heparin-Sepharose chromatography followed by reverse-phase HPLC as previously described (Menozzi *et al*, 1996; Masungi *et al*, 2002).

The human epithelial cell line A549 (ATCC; CCL-185) was cultured in DMEM (GIBCO) containing 10% heat-inactivated FBS, 10 mM HEPES buffer solution, 1 mM sodium pyruvate, and 2 mM ʟ-glutamine. The human monocytic THP-1 cell line (ATCC; TIB-202) was grown in RPMI 1640 (GIBCO), supplemented with 10% (v/v) heat-inactivated fetal calf serum (FCS; GIBCO), 1% (v/v) penicillin–streptomycin (GIBCO), 1% (v/v) sodium pyruvate (GIBCO), 1% (v/v) ʟ-glutamine (GIBCO), and 1% (v/v) HEPES buffer (GIBCO). The THP-1 monocytic cell line was differentiated into macrophages in RPMI medium containing 5 ng/ml phorbol 12-myristate 13-acetate (PMA, Sigma) for 72 h. Primary human airway epithelial

cells isolated from bronchial biopsies (hAECB) were purchased from Epithelix Sàrl and propagated using gamma-irradiated NIH/3T3 cells as feeders (Suprynowicz *et al*, 2012). hAECB cells at 70–80% confluence were separated from feeders by differential trypsinization and passaged to new culture vessels with CnT-Prime Airway medium (CELLnTEC) (A. Imai-Matsushima and T.F. Meyer, manuscript in preparation).

### Flow cytometry

Purified peripheral blood mononuclear cells were stained for 30 min at 4°C with the following panel of mouse anti-human antibodies: CD19 PE-Cy7 (1:25 dilution), IgG V450 (1:20), CD27 FITC (1:25), CD38 PE (1:25), IgA biotin (1:100, all from BD Biosciences) and CD21 APC (1:20; eBioscience). 7AAD (1:200; Life Technologies) was used to exclude dead cells. Biotin-labeled antibodies were detected with streptavidin-Qdot605 (1:100; Life Technologies). BD Bioscience LSRII and Aria I instruments were used for flow cytometric analyses, and Aria II was used for single-cell sorting.

Human lung A549 epithelial cells, human nAECB primary epithelial cells, and human THP-1 monocytes were harvested from culture dishes using either trypsin–EDTA or 3% EDTA and stained with the following anti-human antibodies: CD89 PE (1:10; Biolegend), CD32 PE (1:10; Biolegend), CD16 PE (1:10; BD Biosciences), and CD64 PE (1:10; BD Biosciences). PE-labeled mouse anti-human IgG1 κ and IgG2 κ antibodies (Biolegend) served as isotype controls.

### Single-cell sorting, antibody cloning, and expression

Recombinant monoclonal antibodies were generated as previously described (Tiller *et al*, 2008). In brief, total mRNA was transcribed using random hexamer primers. *IGH* and *IGK* or *IGL* chain transcripts were amplified, sequenced, and cloned into heavy (IgG1, IgA1, or IgA2; Lorin & Mouquet, 2015) or light chain (Igκ or Igλ) expression vectors. Recombinant antibodies were produced in human HEK 293T cells. Recombinant monoclonal IgG or IgA antibodies were purified from culture supernatants using Protein G Fast Flow Sepharose (GE Healthcare) and Protein M Agarose (Invivogen), respectively. Antibodies were eluted from the column by 0.1 M glycine pH 3 and the pH was neutralized with Tris buffer (pH 8). IgG and IgA concentrations were determined by ELISA. Individual batches of final antibody preparations were checked for endotoxin levels with the LAL Chromogenic Endotoxin Quantification Kit (Pierce) following the manufacturer's instructions before use in functional assays and were tested negative for LPS contaminations.

For the functional assays, purified serum antibodies were isolated from plasma using Protein G Fast Flow Sepharose (GE Healthcare) or Protein M Agarose (Invivogen), respectively. Elution and concentration measurements were done as described for monoclonal antibody preparations.

### Sequence analysis

Sequence analyses of the *IGH*, *IGK*, and *IGL* chain transcripts were performed using the IgBlast database to identify germline V(D)J

genes with the highest homology. Somatic hypermutations were determined as previously described (Tiller et al, 2008). The isotype of memory B cells was determined by FACS analysis and the isotype of plasmablast cells by sequence analysis. Sequence and FACS data matched to over 95% when directly compared for memory B cells of TB patient 7.

## ELISA

Polyclonal serum or recombinant monoclonal antibody reactivity was measured as follows. High-binding microtiter plates (Nunc) were coated with 50 μl H37Rv lysate, cell membrane fraction, culture filtrate proteins, or purified native HBHA at a final concentration of 10 μg/ml overnight at 4°C. The plates were blocked with 4% BSA for 1.5 h at room temperature (RT), washed with 0.05% PBS–Tween or PBS, and incubated with diluted plasma or monoclonal antibodies (mAbs) in 0.05% PBS–Tween–1% BSA (plasma) for 2 h or PBS (mAbs) at RT followed by another washing step. HRP-coupled goat anti-human IgG (H+L) antibody at a 1:1000 dilution (Jackson ImmunoResearch) and one-step ATBS substrate (Roche) were used for detection. Area under the curve (AUC) values were calculated with the GraphPad PRISM software. Antibodies were considered MTB reactive when they recognized antigens in the MTB lysate. Antibodies were considered cross-reactive when they recognized at least two out of the four structurally different antigens: MTB lysate, double-stranded DNA, insulin, and LPS. An internal negative control antibody (mGO53; Wardemann et al, 2003) was included in all ELISAs.

Concentration ELISAs were performed as previously described (Tiller et al, 2008). In brief, anti-human Fc-binding IgG (Dianova) was coated overnight at a 1:500 dilution at RT. The plates were blocked with PBS–0.05% Tween for 1 h at RT before incubation with the purified recombinant antibodies for 1.5 h at RT. Human IgG1k from the plasma of myeloma patients (Sigma) was used as standard. Signals were detected as described above.

For LAM ELISAs, MTB Erdman ManLAM and M. smegmatis PILAM were purified and quality control was performed as described previously (Torrelles et al, 2008). Purified LAM (1 μg/well in 0.05 M carbonate–bicarbonate buffer, pH 9.5) was adhered to the wells of an ELISA medium binding plate overnight at 37°C; control wells were left without LAM coating. The wells were washed three times with PBS and blocked with 5% HSA–PBS overnight at 4°C. Blocked wells were washed and incubated with human TB24PB037 antibody, human control antibody (mGO53; Wardemann et al, 2003), or the positive control murine CS-35 antibody in 0.5% HSA–PBS overnight at 4°C. The wells were washed and incubated with HRP-conjugated secondary antibody (1:1,000 dilution; Abcam) in 0.5% HSA–PBS for 2 h at RT. The wells were washed and incubated with HRP substrate (Bio-Rad) at RT for 20 min. The reaction was stopped with 0.2 N sulfuric acid, and the absorbance was measured at 450 nm.

## Infection assay

A549 epithelial cells were grown overnight at $2.5 \times 10^5$ cells/well in 96-well plates. Prior infection, THP-1 monocytic cells ($1 \times 10^6$ cells/well) were differentiated into macrophages as described above. H37Rv bacteria were grown to OD 0.4–0.8, washed with

PBS, and passaged five times through a tuberculin syringe to obtain single-cell suspensions. Recombinant monoclonal antibodies at 10 μg/ml or diluted purified serum IgG/IgA were incubated with $1 \times 10^8$ bacteria for 1 h at 37°C. The cells were washed with FCS-free medium and infected with H37Rv alone or with the antibody–H37Rv mix at an MOI of 1 for 1 h at 37°C. Each condition was tested in triplicate. Cells infected with MTB alone and infected with MTB preincubated with the monoclonal isotype control antibody (mGO53; Wardemann et al, 2003) were used as internal controls. Infected cells were vigorously washed four times with PBS and lysed with 0.5% Triton X-100. Serial dilutions of cell lysates were plated, and the bacterial counts (CFUs) were enumerated at day 21.

## Determination of FcRn expression by Western blot

A549 and THP1 cells were lysed for 5 min on ice with lysis buffer containing 25 mM Tris–HCl pH 7.4, 150 mM NaCl, 1% NP-40, 1 mM EDTA, and 5% glycerol (Pierce). Cell debris was removed by centrifugation, and the cell supernatant was analyzed for FcRn and β-actin protein expression by Western blotting using the following antibodies: anti-human FcRn (1:20 dilution; Novus; NBP1-89128), anti-mouse β-actin (1:1,000; Cell Signaling Technology; 13E5), goat anti-mouse IgG HRP conjugate (1:5,000 dilution; Jackson ImmunoResearch), and goat anti-rabbit HRP conjugate (1:5,000 dilution; Jackson ImmunoResearch).

## Immunohistochemistry and immunofluorescence staining

Lung sections from MTB patients were fixed with 4% paraformaldehyde (PFA) for 1 h and 1% PFA overnight and then embedded in paraffin. Immunostainings were performed after heat-mediated antigen retrieval, using DyLight™594 affinity-purified goat anti-human IgA (10 μg/ml; Jackson ImmunoResearch) and DyLight™649 IgG (10 μg/ml; Jackson ImmunoResearch). Intracellular stainings with polyclonal rabbit anti-human Ki-67/MKI67 antibodies (1:100; Novus) followed by AlexaFluor488-conjugated anti-IgG secondary antibodies (1:100; Jackson ImmunoResearch) were performed after permeabilization with 0.5% Triton X-100.

## Statistics

Statistical significance was determined as described in Appendix Table S5. D'Agostino–Pearson omnibus test was used to determine the normal distribution. Unpaired t-test or Wilcoxon–Mann–Whitney test was used as indicated. Statistical analyses were performed with GraphPad Prism software. A value less than $P < 0.05$ was considered statistically significant (*$P < 0.05$, **$P < 0.01$, ***$P < 0.001$, ****$P < 0.0001$).

## Study approval

All investigations have been conducted according to the Declaration of Helsinki principles. Ethical approval was obtained from Charité University Institutional Review Board (No. EA4/099/11). Written informed consent was received from all participants prior to inclusion in the study. Formalin-fixed, paraffin-embedded lung tissues from three patients with pulmonary MDR-TB were obtained from

---

**The paper explained**

**Problem**

Tuberculosis (TB) caused by *Mycobacterium tuberculosis* (MTB) is a life-threatening infectious disease and a major cause of death worldwide. A vaccine is not available, and antibiotic resistance is becoming an increasing problem. Antibodies against MTB may protect from infection and could be of therapeutic use, but human antibodies against MTB have not been described so far. Here, we set out to identify and characterize human anti-MTB antibodies.

**Results**

Our data show that antibodies against MTB are made in response to acute MTB infection, but also in healthy MTB-exposed individuals. Many of the antibodies were IgA, the dominant antibody isotype produced at mucosal sites, suggesting that they developed during immune responses in the lung. IgA anti-MTB antibodies bound to the surface of MTB and could thereby inhibit the infection of human lung cells. Surprisingly, IgG antibodies, which are prominent during inflammatory conditions including MTB infection, had the opposite effect and promoted the infection. These isotype-dependent differences in host cell infection were also observed with IgA and IgG serum antibodies.

**Impact**

Together, the data provide molecular insights into the human antibody response to MTB and suggest that antibody isotype differences may play a major role in modulating the course of the infection and the development and progression of disease in humans. The targeted induction of IgA antibodies through vaccination may help to protect from MTB infection or promote immunity upon primary exposure.

the Republican Research and Practical Centre for Pulmonology and Tuberculosis (RRPCPT). The use of human biopsies was based on informed patient consent and approved by the ethics commission of the RRPCPT.

**Data availability**

The data reported in this paper are tabulated in Appendix and are archived at GenBank (accession number KX947385–KX949063).

**Expanded View** for this article is available online.

## Acknowledgements

The authors thank Hugo Mouquet for IgA expression vectors, the BEI Resources Repository for reagents, and Simone Kocher for technical assistance. N.Z. was supported by the IMPRS-IDI Berlin. S.H.E.K. acknowledges the support from the European Union's Seventh Framework Programme Project "ADITEC" (HEALTH-F4-2011-280873) and the European Union's Framework Programme for Research and Innovation "TBVAC 2020" (Grant No. 643381). H.W. received funding by the DFG through TRR130 P10.

## Author contributions

NZ designed and conducted the experiments, acquired and analyzed the data, and wrote the manuscript; VT, BH, EA, A-BK, and AI-M conducted the experiments and acquired and analyzed the data; CL and LSS provided reagents and designed the experiments; TZ and MS provided reagents and designed the research study; SHEK and HW designed the research study; and HW wrote the manuscript.

## Conflict of interest

The authors declare that they have no conflict of interest.

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
