## [Review Process File · EMBO Molecular Medicine]

Human isotype-dependent inhibitory antibody responses against *Mycobacterium tuberculosis*

Natalie Zimmermann, Verena Thormann, Bo Hu, Anne-Britta Köhler, Aki Imai- Matsushima, Camille Locht, Eusondia Arnett, Larry S. Schlesinger, Thomas Zoller, Mariana Schürmann, Stefan H.E. Kaufmann, and Hedda Wardemann

Corresponding author: Hedda Wardemann, DKFZ

Review timeline:

Submission date:	23 February 2016
Editorial Decision:	24 March 2016
Revision received:	30 August 2016
Editorial Decision:	06 September 2016
Accepted:	06 September 2016

Transaction Report:

Editor: Céline Carret

1st Editorial Decision

24 March 2016

Thank you for the submission of your manuscript to EMBO Molecular Medicine. We have now heard back from the three referees whom we asked to evaluate your manuscript.

You will see from the reports pasted below, that the referees are overall rather enthusiastic about the study even though referees 1 and 3 do have a few concerns that need to be addressed in a revised version of your article.

Therefore, I'll be happy to consider a revision of your manuscript if you can address the issues that have been raised within 3-months. Please note that it is EMBO Molecular Medicine policy to allow only a single round of revision and that, as acceptance or rejection of the manuscript will depend on another round of review, your responses should be as complete as possible.

I look forward to seeing a revised form of your manuscript as soon as possible.

***** Reviewer's comments *****

Referee #1 (Remarks):

In this study, Zimmermann et al characterized the natural antibody response against TB infection at a molecular and functional level by producing monoclonal antibodies from single plasmablasts and memory B cells in individuals with acute pulmonary TB and in health care workers exposed to

MTB, respectively. The authors found that circulating plasmablasts in patients with acute TB likely differentiated from reactivated "pre-existing" memory B cells. They show that plasmablasts-derived antibodies frequently targeted MTB surface antigens but most importantly, that only IgA antibodies expressed by plasmablasts and memory B cells can inhibit in vitro MTB infection. In contrast, they demonstrate that MTB-specific IgG promoted infection in an Fc receptors dependent manner. These are very important findings that are key in understanding the protective humoral response to TB in humans, and in helping designing therapeutic/vaccine strategies. The study is addressing a research topic of importance to the field, and provides original findings that have important implications for the development of vaccines against TB. The manuscript is very well written, the experiments rigorously performed, and the results based on an impressive amount of work are clearly presented. I have no major concerns but several suggestions for improving the manuscript.

Specific comments:

- 1/ Please provide Tables that were absent from the current submission.
- 2/ Have the authors observed any differences in terms of plasmablasts and memory B-cell antibody responses (as presented in Fig.1) between untreated and treated patients?
- 3/ Can the authors really claim the "low abundance of IgG" (pg7, Fig. 2C) since it appears they constitute about 40-50% in average of the plasmablast compartment. Could they rephrase moderating their statement?
- 4/ Could they mention/comment on the difference of %IgA sequences between TB and HCW (Fig.2J)?
- 5/ The authors tested all their recombinant IgG and IgA mAbs at 10 µg/ml in their in vitro infection assay but it would be really interesting/important to show dose-dependency of the inhibitory activity of IgAs by testing a broader range of concentrations (for few selected IgA but also for IgG antibodies as controls).
- 6/ Concerning the role of Fc receptors (including the FcRn) in potentializing/enhancing TB infection in presence of specific IgGs, the data presented are quite convincing. Nevertheless, I suggest that the authors could test in their infection assay some selected IgG antibodies cloned into a IgH vector carrying mutations decreasing Fc receptors binding such the LALA double mutations (L234A, L235A). These results would indeed strengthen their observation.

Minor comments:

- 1/ Could they provide the values, even if not statistically significant, for the p-value and Spearman correlation "r" in Fig. 1F.
- 2/ To respect the logical order found in the text (pg7 last paragraph), I would suggest swapping panel E and F in Fig.2.
- 3/ Were the frequencies shown in Fig. 3C calculated from binding data on cell lysate plus whole bacteria? Please precise in the legend.
- 4/ Pg10, first sentence: are the results showing the absence of HBHA binding by plasmablasts antibodies presented somewhere, in a Table?

Referee #2 (Comments on Novelty/Model System):

this is a highly relevant study in an important human disease. The technical quality is outstanding.

Referee #2 (Remarks):

The manuscript by Dr. Zimmerman and colleagues represents a unique effort to characterize the antibody response to Mycobacterium tuberculosis in humans. The work is entirely novel and reveals a number of important features of the physiologic response to the pathogen in humans.

Antibodies to mycobacterial surface antigens can be protective in mouse models of TB but whether such antibodies arise normally during human infection is not known. In the first part of the manuscript Zimmerman and colleagues clone antibodies from circulating plasmablasts of infected or

exposed individuals and characterize their activity in vitro. They were able to obtain 230 antibodies from 3 individuals with high titers of anti-TB antibodies including IgG, IgA and IgM antibodies. The mutational load in the plasmablasts indicated that they arose from the memory compartment. The authors prove this point by cloning memory B cells from exposed individuals and showing that anti-HBHA reactive B cells exist in the memory compartment and are comparably mutated.

The cloned antibodies were then tested for reactivity to TB and to specific TB surface antigens. As might be expected many were in fact TB reactive and recognized surface molecules implicated in TB entry into host cells. They then tested whether the antibodies would be protective against infection in a human lung epithelial cell model and found that IgA but not IgG antibodies were protective. They then extended these findings to THP-1 macrophages. Finally they show similar results of purified serum antibodies from the patients.

Overall this is an outstanding piece of work by the leaders in a very difficult area of research.

Referee #3 (Remarks):

In general this is a fine study and this reviewer does not have major criticisms of the work done although I feel that some clarifications are in order (see below). This study analyses antibody function by expressing IgA and IgG from peripheral cells obtained from individual with and without tuberculosis. The experimental work appears to be well done and the manuscript is written clearly. The expressed antibodies were then tested for their efficacy in preventing bacterial infection of human alveolar cell and macrophage lines. The most interesting finding is that the IgA inhibit infection while the IgG facilitate infection. From this data the authors conclude that there are major isotype related differences in the efficacy of antibody against M.tb. Although this statement is almost certainly true there is the important caveat in that the constant region has been shown to affect specificity and one cannot necessarily conclude that the same variable regions cloned into IgG and IgA expression vectors result in the same specificity (for a recent review on this phenomenon see PMID: 26870003). Even if they both bound to the same antigen by ELISA fine specificity can differ. This possibility does not invalidate the finding but does suggest caution.

Specific points

1. The use of an alveolar cell line to study antibody function in vitro is a bit perplexing - are these cells infected in vivo? Some more details should be added as to the rationale for cell selection.
2. I would not draw conclusions about antibody efficacy from THP-1 cells as this is an immortal cell line with unknown relevance to primary cells. The way the text is worded it implies that IgG enhances infection and it may well do so in this cell line by driving phagocytosis without killing. However the situations could be very different in primary cells and there are other studies suggesting that FcR-mediated uptake results in better control of infection. This issue needs to be addressed to avoid adding more confusion to this already confused field.
3. The text uses 'IgG and IgA' but the V regions were cloned into IgG1, IgA1 or IgA2 - which IgA was used in the experiments shown in the figures? The text needs to be amended to state isotype class precisely. The figures should also be modified for clarity.

1st Revision - authors' response

30 August 2016

Referee #1 (Remarks):

In this study, Zimmermann et al characterized the natural antibody response against TB infection at a molecular and functional level by producing monoclonal antibodies from single plasmablasts and memory B cells in individuals with acute pulmonary TB and in health care workers exposed to MTB,

respectively. The authors found that circulating plasmablasts in patients with acute TB likely differentiated from reactivated "pre-existing" memory B cells. They show that plasmablasts-derived antibodies frequently targeted MTB surface antigens but most importantly, that only IgA antibodies expressed by plasmablasts and memory B cells can inhibit *in vitro* MTB infection. In contrast, they demonstrate that MTB-specific IgG promoted infection in an Fc γ 3 receptors dependent manner. These are very important findings that are key in understanding the protective humoral response to TB in humans, and in helping designing therapeutic/vaccine strategies. The study is addressing a research topic of importance to the field, and provides original findings that have important implications for the development of vaccines against TB. The manuscript is very well written, the experiments rigorously performed, and the results based on an impressive amount of work are clearly presented. I have no major concerns but several suggestions for improving the manuscript.

We would like to thank the reviewer for his positive comments and suggestions to improve the manuscript and are happy that he/she acknowledges the importance and originality of our findings.

Specific comments:

1/ Please provide Tables that were absent from the current submission.

We have now included the tables in the appendix as tables S1-S4 and apologize that they were missing from the original submission.

2/ Have the authors observed any differences in terms of plasmablasts and memory B-cell antibody responses (as presented in Fig.1) between untreated and treated patients?

We did not observe any differences in the serum antibody response between untreated and treated patients. The information is now provided in Table I. As stated on p6 of the manuscript in individual donors "the frequency of plasmablasts was highest during early acute TB and waned upon drug-treatment, whereas serum IgG levels increased over 6 months of antibiotic combination therapy (Fig 1E). Several untreated patients with prominent serum antibody responses lacked detectable plasmablast levels in the circulation suggesting that these donors had been infected for prolonged times so that their circulating plasmablast response had waned (Fig 1F)."

3/ Can the authors really claim the "low abundance of IgG" (pg7, Fig. 2C) since it appears they constitute about 40-50% in average of the plasmablast compartment. Could they rephrase moderating their statement?

We agree with the reviewer that the statement was too strong and have modified the sentence on p7 accordingly: "The relative bias towards IgA and near complete absence of IgM expression compared to circulating memory B cells from the same donors indicated a mucosal origin (Fig 2C).

4/ Could they mention/comment on the difference of %IgA sequences between TB and HCW (Fig.2J)?

We have modified the sentence describing Figure 2J as follows:

Results (p8): „The low frequency of IgG was more pronounced in HCW than in TB patients whereas IgA was particularly more abundant in HCW suggesting an association of disease onset with induction of IgG responses (Fig 2J). ... The high frequency of IgA anti-HBHA memory B cells in HCW suggests that memory is formed upon primary MTB exposure presumably from mucosal immune responses. Active TB could lead to reactivation of pre-existing memory B cells and formation of plasmablast responses that are associated with class-switching to IgG.”

We would like to draw the reviewer's attention to the discussion (p14) where we interpret the data: „However, IgG responses develop particularly under inflammatory conditions and anti-MTB IgG serum levels are higher in patients with active disease compared to HCW (Demkow et al, 2007; Pukazhvanthen et al, 2014)...How antibody isotype differences influence the course of natural MTB infection in humans and whether isotype switching precedes the development of active TB or reflects a consequence of disease remains to be determined...”

5/ The authors tested all their recombinant IgG and IgA mAbs at 10 µg/ml in their in vitro infection assay but it would be really interesting/important to show dose-dependency of the inhibitory activity of IgAs by testing a broader range of concentrations (for few selected IgA but also for IgG antibodies as controls).

To address this point we have tested a few IgA and IgG antibodies at 100, 10 and 1 µg/ml. As expected, the infection promoting and inhibitory activity of IgG and IgA antibodies, respectively, was concentration-dependent. We now added this data as Fig S1 in the Appendix and comment on the results in the results section on p11/p12. Differences in the degree of concentration-dependency may depend on the antigen-specificity and affinity.

6/ Concerning the role of Fcγ receptors (including the FcRn) in potentializing/enhancing TB infection in presence of specific IgGs, the data presented are quite convincing. Nevertheless, I suggest that the authors could test in their infection assay some selected IgG antibodies cloned into a IgH vector carrying mutations decreasing Fcγ receptors binding such the LALA double mutations (L234A, L235A). These results would indeed strengthen their observation.

Reviewer 3 was concerned that THP-1 cells may not reflect what could be seen with primary macrophages. We therefore focused the manuscript on the infection of epithelial cells (please see response to reviewer 3 below). THP-1 cells were used as positive control for the analysis of IgA and IgG Fc receptor expression and to illustrate that IgA and IgG anti-MTB antibodies showed the same effect on MTB infection in the presence of these receptors. Any discussion on the potential effect of the antibodies on macrophages was removed from the manuscript since we agree with reviewer 3 that THP-1 cells cannot predict what would be seen with primary macrophages. Unfortunately we could not address this point experimentally since we did not get access to primary human alveolar macrophages, particularly not from BAL of healthy individuals not treated with immunosuppressive drugs.

Since the LALA mutations would affect the interaction of IgG with conventional Fcγ receptors and not the interaction with FcRn, we did not test LALA mutated IgG antibodies in our A549 infection model as the cells lack conventional FcR and only expressed FcRn. We hope that the reviewer agrees to the modified version of the manuscript and the focus on epithelial cells.

Minor comments:

1/ Could they provide the values, even if not statistically significant, for the p-value and Spearman correlation "r" in Fig. 1F.

The p-value and Spearman correlation "r" are now indicated in Fig. 1F.

2/ To respect the logical order found in the text (pg7 last paragraph), I would suggest swapping panel E and F in Fig.2.

We swapped the panels as suggested.

3/ Were the frequencies shown in Fig. 3C calculated from binding data on cell lysate plus whole bacteria? Please precise in the legend.

This is correct. The frequencies were calculated from binding data obtained in the cell lysate and whole bacteria assay. We now specify this in the legend of Fig. 3C p33: "...Pie charts show frequency of MTB-reactive (grey) vs. non-reactive (white) antibodies as measured by whole cell lysate and MTB bacteria ELISA for each patient. MTB-reactive antibodies were positive in both or one of the assays..."

4/ Pg10, first sentence: are the results showing the absence of HBHA binding by plasmablasts antibodies presented somewhere, in a Table?

We apologize. The negative data was not shown. We now state this in the text on p10: "...Although none of the plasmablast antibodies recognized HBHA (data not shown), the detection of HBHA-reactive memory B cells in HCW and TB patients with acute disease (Fig 2D-I)..."

Referee #2:

This is a highly relevant study in an important human disease. The technical quality is outstanding.

The manuscript by Dr. Zimmerman and colleagues represents a unique effort to characterize the antibody response to Mycobacterium tuberculosis in humans. The work is entirely novel and reveals a number of important features of the physiologic response to the pathogen in humans.

Antibodies to mycobacterial surface antigens can be protective in mouse models of TB but whether such antibodies arise normally during human infection is not known. In the first part of the manuscript Zimmerman and colleagues clone antibodies from circulating plasmablasts of infected or exposed individuals and characterize their activity in vitro. They were able to obtain 230 antibodies from 3 individuals with high titers of anti-TB antibodies including IgG, IgA and IgM antibodies. The mutational load in the plasmablasts indicated that they arose from the memory compartment. The authors prove this point by cloning memory B cells from exposed individuals and showing that anti-HBHA reactive B cells exist in the memory compartment and are comparably mutated.

The cloned antibodies were then tested for reactivity to TB and to specific TB surface antigens. As might be expected many were in fact TB reactive and recognized surface molecules implicated in TB entry into host cells. They then tested whether the antibodies would be protective against infection in a human lung epithelial cell model and found that IgA but not IgG antibodies were protective. They then extended these findings to THP-1 macrophages. Finally they show similar results of purified serum antibodies from the patients.

Overall this is an outstanding piece of work by the leaders in a very difficult area of research.

Referee #3 (Remarks):

In general this is a fine study and this reviewer does not have major criticisms of the work done although I feel that some clarifications are in order (see below). This study analyses antibody function by expressing IgA and IgG from peripheral cells obtained from individual with and without tuberculosis. The experimental work appears to be well done and the manuscript is written clearly. The expressed antibodies were then tested for their efficacy in preventing bacterial infection of human alveolar cell and macrophage lines. The most interesting finding is that the IgA inhibit infection while the IgG facilitate infection. From this data the authors conclude that there are major isotype related differences in the efficacy of antibody against M.tb. Although this statement is almost certainly true there is the important caveat in that the constant region has been shown to affect specificity and one cannot necessarily conclude that the same variable regions cloned into IgG and IgA

expression vectors result in the same specificity (for a recent review on this phenomenon see PMID: 26870003). Even if they both bound to the same antigen by ELISA fine specificity can differ. This possibility does not invalidate the finding but does suggest caution.

We agree with the reviewer that antibody isotype may affect specificity. To address this point we have performed a direct comparison of two representative anti-HBHA IgA and IgG antibodies including ab HCW1HBMem095, which showed comparable ELISA reactivity but significant differences in its functional activity as shown in Fig. 5E. The data is now included in Fig. EV8.

Specific points

1. The use of an alveolar cell line to study antibody function in vitro is a bit perplexing - are these cells infected in vivo? Some more details should be added as to the rationale for cell selection.

We have included a sentence and references to the results section on p10 to justify the choice of the cell line. A549 cells are Type II alveolar epithelial cells and MTB has been shown to invade and replicate in the cells. We would like to refer the reviewer to the following literature:

- Castro-Garza J, King CH, Swords WE, Quinn FD (2002) Demonstration of spread by *Mycobacterium tuberculosis* bacilli in A549 epithelial cell monolayers. *FEMS Microbiol Lett* 212: 145-149
- Bermudez, L. E. & Goodman, J. (1996). *Mycobacterium tuberculosis* invades and replicates within type II alveolar cells. *Infection and Immunity* 64, 1400–6.
- Sato, K., Tomioka, H., Shimizu, T. et al. (2002). Type II alveolar cells play roles in macrophage-mediated host innate resistance to pulmonary mycobacterial infections by producing proinflammatory cytokines. *Journal of Infectious Diseases* 185, 1139–47.
- McDonough, K. A. & Kress, Y. (1995). Cytotoxicity for lung epithelial cells is a virulence-associated phenotype of *Mycobacterium tuberculosis*. *Infection and Immunity* 63, 4802–11.
- Hernandez-Pando, R., Jeyanathan M., Mengistu G., Aguilar D., Orozco H., Harboe, M., et al. (2000) Persistence of DNA from *Mycobacterium tuberculosis* in superficially normal lung tissue during latent infection. *The Lancet* 356, 2133–8.
- Birkness, K.A., Deslauriers, M., Bartlett, J.H., White, E.H., King, C.H., Quinn, F.D. (1999) An In Vitro Tissue Culture Bilayer Model To Examine Early Events in *Mycobacterium tuberculosis* Infection. *Infection and Immunity* 67, 653–8.
- Ryndak, M.B., Singh, K.K., Peng, Z., Laal, S. (2015) Transcriptional Profile of *Mycobacterium tuberculosis* Replicating in Type II Alveolar Epithelial Cells. *PLOS One* 10:e0123745.
- Zimmermann N., Saiga H., Houthuys E., Moura-Alves P., Koehler A., Bandermann S., Dorhoi A., Kaufmann SH. (2016) Syndecans promote mycobacterial internalization by lung epithelial cells. *Cell Microbiol*. doi: 10.1111/cmi.12627.

2. I would not draw conclusions about antibody efficacy from THP-1 cells as this is an immortal cell line with unknown relevance to primary cells. The way the text is worded it implies that IgG enhances infection and it may well do so in this cell line by driving phagocytosis without killing. However the situations could be very different in primary cells and there are other studies suggesting that FcR-mediated uptake results in better control of infection. This issue needs to be addressed to avoid adding more confusion to this already confused field.

We agree with the Reviewer that the relevance of the THP-1 data to primary cells is unclear. Unfortunately we could not address this point experimentally since we did not get access to primary human alveolar macrophages, particularly not from BAL of healthy individuals not treated with immunosuppressive drugs. We therefore decided to focus the manuscript on the infection of epithelial cells and changed Fig. 6 and the wording of the manuscript on p12 to avoid any confusion.

3. The text uses 'IgG and IgA' but the V regions were cloned into IgG1, IgA1 or IgA2 - which IgA was used in the experiments shown in the figures? The text needs to be amended to state isotype class precisely. The figures should also be modified for clarity.

We apologize that the isotype subclass was not indicated. The original isotype is given in the Tables II-IV, which unfortunately were not included in the first version of the submitted manuscript. We have now also indicated the isotype subclass information in the respective figures and legends.

2nd Editorial Decision

06 September 2016

Please find enclosed the final reports on your manuscript. We are pleased to inform you that your manuscript is accepted for publication and will be sent to our publisher to be included in the next available issue of EMBO Molecular Medicine. Please send us the missing accession numbers if you have obtained them or make sure to supply them to production as soon as you do have them.

Congratulations on your interesting work.

***** Reviewer's comments *****

Referee #1 (Remarks):

The authors have appropriately addressed all my initial concerns/comments, and the manuscript has been considerably improved by the revisions, which include new experiments/data. I strongly consider that the revised version now provided by the authors fits the quality requirements for publication in EMBO Molecular Medicine, and that the study will be of major interest to its readers, and more broadly to the scientific community.

Referee #3 (Comments on Novelty/Model System):

no additional suggestions

Referee #3 (Remarks):

The authors have addressed my concerns and I have no additional comments

Corresponding Author Name: Hedda Wardemann

Manuscript Number: EMM-2016-06330